# Addition of Graphite Filler to Enhance Electrical, Morphological, Thermal, and Mechanical Properties in Poly (Ethylene Terephthalate): Experimental Characterization and Material Modeling

**DOI:** 10.3390/polym11091411

**Published:** 2019-08-28

**Authors:** Basheer A. Alshammari, Fahad S. Al-Mubaddel, Mohammad Rezaul Karim, Mokarram Hossain, Abdullah S. Al-Mutairi, Arthur N. Wilkinson

**Affiliations:** 1Materials Research Institute, King Abdulaziz City of Science and Technology, Riyadh 11442, Saudi Arabia; 2Chemical Engineering Department, King Saud University, Riyadh 11421, Saudi Arabia; 3Center of Excellence for Research in Engineering Materials, King Saud University, Riyadh 11421, Saudi Arabia; 4Zienkiewicz Centre for Computational Engineering, College of Engineering, Swansea University, Bay Campus, Swansea SA1 8EN, UK; 5School of Materials, The University of Manchester, Manchester M13 9PL, UK

**Keywords:** graphite, PET, conductive fillers, electrical, morphological, thermal properties, filled polymer model, large strain model

## Abstract

Poly(ethylene terephthalate)/graphite (PET/G) micro-composites were fabricated by the melt compounding method using a minilab extruder. The carbon fillers were found to act as nucleating agents for the PET matrix and hence accelerated crystallization and increased the degree of crystallinity. TGA showed that carbon fillers improved the resistance to thermal and thermo-oxidative degradation under both air and nitrogen atmospheres. However, a poor agreement was observed at higher loadings of the filler where the composites displayed reduced reinforcement efficiency. The results demonstrate that the addition of graphite at loading >14.5 wt.% made electrically conductive composites. It was calculated that the electric conductivities of PET/graphite micro-composites were enhanced, above the percolation threshold values by two orders of magnitudes compared to the PET matrix. The minimum value of conductivity required to avoid electrostatic charge application of an insulating polymer was achieved, just above the threshold values. The addition of graphite also improved thermal stability of PET, accelerated its crystallization process and increased the degree of crystallinity. Microscopic results exhibit no indication of aggregations at 2 wt.% graphite, whereas more agglomeration and rolling up could be seen as the graphite content was increased in the PET matrix (in particular, above the percolation threshold value). Furthermore, based on the mechanical experimental characterization of the PET/graphite micro-composites, a large deformation-based mathematical model is proposed for material behavior predictions. The model fits well the experimental data and predicts other mechanical data that are not included in the parameter identification.

## 1. Introduction

Much research have been focused on preparing polymeric matrix composites (PMC) for high performance applications, particularly using carbon and its allotropes, which include both micro- and nano-size fillers such as carbon black (CB), graphite, carbon fibers, graphite nanoplatelets (GNP), graphene and carbon nanotubes (CNT) [1,2,3,4]. Traditionally, polymers are filled with micro-fillers to improve electrical and mechanical properties. However, a high loading is required which can negatively influence mechanical properties and processing [2,3]. In contrast, earlier studies have demonstrated that even low addition levels of nano-fillers can give significant improvements to the electrical, mechanical, and thermal properties of polymers [4,5,6]. Furthermore, the nano-fillers can also improve the flame retardant properties of polymers (see, for example, [7,8]). Nevertheless, poor interfacial adhesion between the reinforcement and the matrix, inadequate dispersion and non-uniform distribution are parameters of major concern that need to be addressed before reaping the full potentials of particulate conductive nano-fillers. To resolve these issues, sizeable efforts have been also applied to the chemical modification of nano-fillers [9].

Micromechanics show that the properties of polymer composites are a function of the behavior of the individual constituents, their shapes and arrangements, volume fractions, and the interfaces between matrices and reinforcements [2]. For the micro-scale fillers, the properties are largely independent of their sizes. However, when reinforcements are of nano-scale, their size plays a vital role due to their high surface areas and aspect ratios, as in the case of CNT, graphene, and GNP. The most common applications of conductive polymer/carbon composites are as antistatic and electromagnetic shielding materials. However, these composites could also be used as heaters, separators or electrodes [10,11,12,13]. Despite a wide range of applications, there is a lack of understanding of the processing–structure–property correlations for these composites. In particular, comparative studies dealing with various types of carbon fillers incorporated into polymers [13] have not been well-studied. Therefore, the aim of the present work was to prepare conductive micro-composites using poly(ethylene terephthalate) (PET) matrix and carbon fillers such as graphite to investigate their processing–structure–property relationships. PET is a thermoplastic semi-crystalline polymer used widely in applications such as fibers, films and packaging thanks to its good strength, chemical resistance, and dimensional stability [14,15]. Despite the aforementioned properties of PET, improvements of its electrical, thermal and mechanical properties are required for high performance applications which were the objectives of the present study. An improved electrical conductivity of PET is needed for making electrostatic devices [16]. Another exciting application of conductive PET/carbon nano-composites is the replacement of indium tin oxide (ITO) electrodes because of the latter poor mechanical and higher sheet resistance when compared with CNT [17,18].

The most commonly used and extensively researched particulate carbon fillers and various types of polymers can be used as matrices to produce PMC including thermoplastics, thermosetting polymers, and elastomers [19,20,21,22]. Graphite is a micron-scale carbon filler that is commonly used in the fabrication of polymer composites [21,22,23,24,25]. It exists naturally and can also be synthetically prepared [23,26]. The elemental carbon is at its lowest energy level in graphite at room temperature [13,27]. The structure of graphite consists of parallel layers of graphene sheets with ${\mathrm{sp}}^{2}$-hybridized carbon that is bonded hexagonally [13]. In graphite, carbon atoms are connected to each other through strong covalent bonds within the graphene sheets, whereas the parallel sheets are held together by weak van der Waals forces of attraction. Thus, graphite is an anisotropic material whose elastic modulus is significantly higher in parallel (1 TPa) but lower in the perpendicular direction (36.4 GPa) [26]. Graphite has high elastic modulus, thermal conductivity, thermal stability, and good electrical resistant (∼50μΩcm at room temperature) [13,23,27,28]. It should be noted that the π orbitals spread into entire graphene layers of graphite allowing electrical and thermal conductivities [13]. It is often difficult to produce satisfactory polymer nano-composites as several studied have noted [24,29,30]. There is no single unanimously agreed production method, as each type of polymer may need special processing techniques resulting in wide variations in properties [2,18]. The environmentally friendly mass production route known as melt compounding was employed in this research. This is a common method of preparing polymer–particulate composites [13]. Hence, it is usual to investigate this method for producing polymer/carbon composites.

The properties of polymer composites depend on sizes and dimensions of fillers. Micro-fillers are used to fabricate composites due to their good mechanical properties, lower costs, and easy availability. Therefore, in this study, conductive carbon filled PET composites were prepared by using graphite to study the electrical conductivity, morphology, and thermal properties of PET matrix and its micro-composites. At first, the percolation threshold ($\Phi_{c}$) value was determined by using the percolation theory as this value is very important for improving fabrication of the polymer composites (PC) and their properties. The development of PC based on conductive fillers has concentrated on reducing concentration of fillers (Φ) in order to reduce cost, improve the fabrication or processability, and enhance thermal properties of the final composite products [31]. The present study was designed to fabricate micro-composites using PET as the matrix and graphite as the conductive carbon filler by the minilab extruder, with an aim to investigate their processing parameters and structure–property correlations. The electrical, morphological, thermal, and mechanical behavior of these composites were investigated in this study.

Constitutive modeling of polymeric materials and their composites help get insights into material mechanical responses. To predict mechanical responses under the influence of micro-scale fillers, mathematical models for computer-based simulations have increasingly been developed in the literature [32,33,34]. Within very small mechanical deformations (less than one percent), small strain-based models have been used [32,33]. For large deformations, some advanced models for particle-filled polymers, which are based on the so-called finite strain theory, can be found in the literature [34]. After synthesizing and extensively characterizing the graphite enhanced PET in our study, a comprehensive mechanical characterization was conducted, i.e., stress responses were calculated with different deformations up to the material break. As expected, the rigid graphite filler stiffens PET polymer. After a set of mechanical experiments, a large deformation-based material model is proposed. For material parameters identification, a stress–stretch dataset was used. For the model prediction, another dataset, was not used during parameter identification, was utilized. In this case, large deformation-based models can predict stress–stretch responses quite satisfactorily.

The paper is organized as follows. In Section 2, a brief description is presented outlining the material preparations and characterization techniques in our laboratory. In Section 3, detailed results and corresponding discussions are presented. In these cases, morphological, electrical, mechanical, and thermal property characterization and significant enhancements of the PET/graphite micro-composites are analyzed. In an effort to develop a material model, mechanical characterizations and modeling of filled polymers are elaborated in Section 4. Therein, the constitutive equations are formulated in one-dimensional form suitable for the parameter identification as well as for the model validation. In Section 4, the validation of the model with the experimental data is demonstrated and discussed. Finally, Section 5 presents a brief summary of the current work.

## 2. Experimental Details

### 2.1. Materials

PET was received as a “resin” and purchased from Equipolymers (grade: LIGHTER C93). Note that, to be consistent with the company’s (Equipolymers) manual [35], we here loosely use the term “resin”. As-received flakes of synthetic graphite were purchased from Sigma-Aldrich in the form of powder with a particle size of <20μm and density of 2.26 g/cm${}^{3}$.

### 2.2. Preparation of PET/Graphite Micro-Composites

Melt compounding technique was used for the preparation of PET/graphite micro-composites. The PET and the graphite filler were dried in a vacuum oven at 120 °C for 24 h prior to the melt compounding. The micro-composites were mixed by melt compounding using a laboratory scale (7 cm${}^{3}$) Thermo-Haake minilab co-rotating twin-screw compounder. The operational conditions were 5 min mixing time with a temperature of 270 °C at a screw speed of 45 rpm. The extruded samples were cooled by passing through an ice-water bath, chopped into pellets using a pelletizer and then dried before any further processing by compression or injection moldings. Extruded PET and the composite samples were compression molded to obtain films of about 1 mm thick (frame mold). The compression molding procedure involved preheating at 280 °C for 10 min, followed by compression for an additional 10 min at a pressure of 18 MPa at the same temperature. This was followed by quenching in an ice-water bath and drying in the vacuum oven at 40 °C for 24 h. The quenched films were stored for further characterization and analytical tests. The degree of crystallinity of the PET matrix can be controlled by the cooling rate of the melt. Therefore, the effect of cooling rate on the crystallinity of PET was investigated. These studies are discussed in more detail in Section 3.

### 2.3. Characterization of Micro-Composites

The electrical conductivities of the PET/graphite micro-composites were measured at room temperature using a phase sensitive multimeter (NumetriQ PSM1735). Each sample was examined five times to check the result accuracy. The morphology state of graphite in the micro-composites was characterized by using a scanning electron microscope (Philips SEM XL30), at an accelerating voltage of 10–20 kV. For SEM images of the micro-composites, their films were cryogenically fractured in a liquid nitrogen bath. After that, samples were mounted on 0.5-inch pin stubs using a carbon adhesive tape and then coated with thin layer gold using an Edwards S150B sputter coater, to prevent any charging. The samples were chosen to examine their morphologies below, close, and above the percolation threshold value of PET/graphite composite system. Melting and crystallization behavior of the micro-composites were examined by differential scanning calorimetry (TA Instrument DSC Q-100). All measurements were scanned from room temperature to 270 °C and the test was performed under nitrogen gas condition using a heat–cool–heat run at a heating/cooling rate of 10 °C/min. To confirm accuracy of the results, three samples from each material were measured. The thermal stability of pure PET and its micro-composites was investigated as a function of graphite contents by using a thermogravimetric analyzer (TGA, TA Q-500). All samples (≈5 mg) were scanned from 23 to 700 °C under nitrogen and air gas condition at a heating rate of 10 °C/min.

## 3. Results and Discussions

### 3.1. Morphological Characterization of PET/Graphite Micro-Composites

The as-received graphite was examined using the SEM to study their morphology before incorporation into the PET matrix. The graphite exhibits large platelets of the scale of ∼20–100 μm as, shown in Figure 1.

Figure 2 shows SEM micrographs of the PET/graphite micro-composites. The graphite flakes appear white in the images and the PET matrix is seen as grey. At the loading of graphite below the percolation threshold value ($\Phi_{c}$ = 14.7), graphite particles are relatively far from each other (Figure 2a). As a result, the electrons cannot move effectively. Thus, the resistance of the micro-composites remains high. However, when the graphite loading reaches $\Phi_{c}$ (Figure 2b), enough particles are exposed to each other to form a conductive network. This transforms the insulating PET into a conductive material. When the graphite loading is increased above the percolation threshold, an agglomeration is observed and some graphite layers are sufficiently large to be pulled out of PET matrix during fracture (Figure 2c). Moreover, the PET/graphite micro-composites consist of both large and small particles. The small particles could have been formed during the melt mixing process, which causes fragmentation of the large particles.

### 3.2. Electrical Properties of PET/Graphite Micro-Composites

From the impedance spectroscopy, the real and imaginary parts of the complex impedance ertr obtained as a function of frequency. Values for the conductivities were determined from the real part of the complex impedance using the following equation [36,37]:(1)$$\begin{array}{c} \sigma =\frac{L}{RA}, \end{array}$$
where σ is the electrical conductivity (S/m), L is the distance between electrodes (*m*), A is the cross sectional area of the sample ($m^{2}$) and R is the measured electrical resistance (Ω). In Figure 3, it is clear that a large increase in σ occurs between 14 wt.% and 15 wt.%. Therefore, the percolation threshold value $\Phi_{c}$, defined as the minimum loading of the filler associated with the change in the electrical behavior of the composite from insulated to conductive, occurs between 14 wt.% and 15 wt.% for the PET/graphite micro-composites. The theoretical electric conductivity curve for composites indicates that the PET/graphite composites exhibit a typical percolation transition. In general, σ increases as the graphite loading increases up to a certain loading, after which only moderate increases in σ are observed with a further addition of graphite. The addition of about 14.7 wt.% of graphite increases σ by ∼2 orders of magnitude at 10 Hz compared to the pure PET matrix. The value of σ is about $10^{-6}$ S/m, which is considered to be the minimum value for avoidance of electrostatic charge build-up in an insulating matrix [37]. The maximum value of σ of the PET/graphite composites is 0.0016 S/m at 25 wt.%, which is within the range for semiconducting materials [38]. At this threshold, the variation in the conductivity of a polymer composite as a function of the conductive filler content exhibits usually an S-shaped curve, which can be described by a power-law relationship according to the percolation theory [39]. The theory is often used to characterize the insulator–conductor transitions of polymer composites containing conductive fillers in order to determine the percolation threshold value and the dimensionality of the conductive path network in polymer matrices.

According to the percolation theory [36,37,38,39,40,41], the following equation gives the value of σ of the polymer/carbon composites:(2)$$\begin{array}{c} \sigma =\sigma_{0}{\left[\Phi -\Phi_{c}\right]}^{t},\;\mathrm{for}\;\Phi >\Phi_{c} \end{array}$$
where σ is the conductivity of composite (S/m), $\sigma_{0}$ is the proportionality constant, $\Phi_{c}$ is the critical loading (percolation threshold in vol.%), Φ is the filler content (vol.%), and *t* describes the dimensionality of the system that depends on the geometry of the network. In theory, *t* of 1.3 and 2.2 represent two- and three-dimensional systems, respectively. However, experimental values outside this range are also reported in the literature [38]. The fitting of the percolation equation to the experimental data is represented in Figure 3 (insert graph) for PET/graphite micro-composites. Correlation factor $R^{2}$ was very close to unity, demonstrating a good fit of the experimental data to the power-law model. A best fit to the data was achieved at Φ = 14.7 wt.% of graphite and t=1.3. Hence, the PET/graphite micro-composites showed percolation threshold value $\Phi_{c}$ = 14.7 wt.% in this study. Krupa et al. [29] examined electrical properties of PE/graphite composites and reported even higher $\Phi_{c}$ value of ∼11 vol.% (∼24 wt.%). She et al. [42] also reported higher $\Phi_{c}$ values, i.e., ∼22.2 wt.% for PE/graphite micro-composites. Therefore, the values available in the literature are higher in comparison to those obtained in the present study. The difference in percolation threshold values could be attributed to the difference in filler sizes and preparation method used. Note that the inclusion of a stiff and conductive graphite filler increases the mechanical stiffness of PET composites. which has disadvantages (e.g., a reduction of stretchability and increase of embrittlements) in many areas, e.g., dielectric elastomers for soft robotics [43,44]. Therefore, the synthesis of an optimized composite with a minimum percolation threshold mostly has positive outcomes.

### 3.3. Thermal Analyses of PET/Graphite Micro-Composites

One of the objectives of performing thermal analyses is to obtain thermodynamic properties. Thermodynamics denotes the motion of energy on all levels. These properties control the rate of energy exchange and absorbance from an energy source. Properties such as specific heat and latent heats of phase change are essential parameters in the study of thermal behavior and stability of polymer composites. The TG/DTA Thermal analyzer used in this study gives a DSC signal as well. The DSC study shows the rate and magnitude of energy changes experienced by the sample. From the DSC curve, properties such as specific heat, latent heats of phase change or transition, and temperature of transition can be obtained directly.

#### 3.3.1. Crystallization Behavior

The pure PET and its micro-composites with varying concentration of synthetic graphite (2, 5, 10, 15, and 20 wt.%) were subjected to a heat–cool–heat cycle at 10 °C/min in the DSC. The DSC curves in Figure 4a–d show a range of peaks: cold crystallization peaks ($T_{cc}$), melting peaks from the first heating scans ($T_{m}$), melt crystallization peaks during the cooling ($T_{mc}$) and the melting peaks from second heating scans of the composites. The data derived from these curves are summarized in Table 1. The incorporation of graphite into the PET matrix is seen to significantly affect crystallization behavior. As shown in Figure 4a,c and Table 1, the cold crystallization temperature values ($T_{cc}$) decrease with an increase of graphite in the PET matrix, indicating that the cold crystallization of PET in the graphite composites is nucleated at a lower temperature than in the pure PET matrix. Moreover, the melt crystallization temperature ($T_{mc}$) of the composites increased from ∼209 to 223 °C as the graphite content increased from 0 to 20 wt.%. The degree of super-cooling (ΔT) decreased with increasing of graphite concentration, from ∼44 °C for pure PET to ∼36 °C for PET composites containing 2 wt.% of graphite content and to ∼30 °C for those with 20 wt.%. All observations here indicate clearly that graphite acts as a strong nucleating agent, ascribed to the fact that the filler is well known to interact with PET repeat units [26,45], causing restrictions in their movement and ultimately resulting in heterogeneous nucleation. Incorporation of >2 wt.% graphite results only in a relatively small shift in the crystallization temperatures. It is revealed in this study that 2 wt.% loading of the graphite is the optimum loading for the filler material to act as a nucleating agent for the PET matrix.

Typically, an efficient nucleating agent reduces the energy required for nucleation and hence accelerates the crystallization process [46]. The heterogeneous nucleation of PET with carbon fillers is well known, and has been reported for the carbon-based fillers such as CNT, graphite, GNP, and carbon black (CB) [31,46,47,48]. For example, Xin et al. [31] reported an increase of 11 °C in the $T_{mc}$ of pure PET with a loading of 5 wt.% graphite. Moreover, the percentage of the degree of crystallinity ($X_{c}$) of PET and its carbon composite was calculated from the first-heat data as these data reflect the thermal history of the PET/carbon composites. The initial $X_{c}$ (of the specimen placed in the DSC) can be calculated using the enthalpies of both crystallization and melting, according the following equation [48,49]:(3)$$\begin{array}{c} X_{c}=\frac{\Delta H_{m}-\Delta H_{cc}}{\left[1-v_{f}\right]\Delta H_{0}}\times 100, \end{array}$$
where $\Delta H_{m}$ is the melting enthalpy (J/g) measured in the heating experiment, $\Delta H_{cc}$ is the cold crystallization enthalpy (J/g), $\Delta H_{0}$ is the theoretical enthalpy of 100% crystalline PET ($\Delta H_{0}$ = 140 J/g) [50,51] and $v_{f}$ is the weight fraction of carbon fillers.

The nucleation process is also reflected in the $X_{c}$ values obtained after quenching (Table 1). Values of $X_{c}$ were calculated from the first heating cycles using Equation (Equation 3) as these values reflect the thermal history of the PET/graphite micro-composites. $X_{c}$ was found to increase upon adding graphite; from 12% to 16% at addition levels up to 10 wt.%, rising to ≈20% upon addition of 15–20 wt.% of graphite. Similar enhancements in $X_{c}$ have been reported for poly(vinylidene fluoride) (PVDF)/graphite composites [25], PP/graphite composites [24] and PET/graphite composites [31]. However, Xin et al. [31] found that the $X_{c}$ of PET decreased when the loading of graphite exceeded 10 wt.%. They attributed this decrease to the barrier effect of graphite, via which graphite obstructs crystal growth. Incorporating graphite into the PET matrix seems to have no appreciable effect on the melt temperature ($T_{m}$), which stays essentially constant at 254 ± 1 °C (Figure 4b,d and Table 1). Similar results have been reported for PET/carbon composites; such as PET/GNP nano-composites [48], PET/multi-wall CNT nano-composites [52,53,54] and PET/single-wall CNT nano-composites [55,56]. In addition, the values of glass transition temperature ($T_{g}$) were found to be unaffected by the incorporation of graphite up to 10 wt.%. However, at 15 wt.%, which is above the percolation threshold, $T_{g}$ values drop by nearly 8 °C. The $T_{g}$ of polymers was found to be dependent on the free volume that is available for the movement of polymer chains [40,57]. The free volume has a critical value which defines the $T_{g}$, because this facilitates the chains segmental motion. Thus, a high graphite content may have influences on the free volume that is an indicative of reduction in $T_{g}$ of composites. However, interactions between carbon fillers and a PET matrix have been reported to give a decrease [53,57], increase [58] or no change [47] in the $T_{g}$ of the composites. The reduction in the $T_{g}$ values of the PET/graphite composites could also be attributed to a poor affinity of graphite for the PET matrix, as its loading is increased [45].

Graphite agglomerates were observed in the PET matrix at higher loadings using SEM images, as shown in Figure 2c. A reduction in $T_{g}$ upon addition of 10 wt.% of expanded graphite into a phenylethyny-terminated polyimide matrix (PETI-5) was reported, which was attributed to a poor dispersion (partially agglomerated) in these PETI-5/graphite micro-composites [59].

#### 3.3.2. Thermal Stability of the Micro-Composites

Figure 5 shows TGA results collected for all the PET/graphite micro-composites under nitrogen and air atmospheres compared to the neat PET matrix. The results demonstrate no significant weight loss up to ∼350 °C for all PET composites. As the temperature was increased (>350 °C), the weight loss increased significantly over a narrow temperature range, as seen by the steep slopes. However, the onset temperature at which the weight loss begins is different, as shown by the inset figures. The degradation temperature at 5% of mass loss (°C), i.e., $T_{5\%}$, for each of composites is reported in Table 2. The mass of residue for each composite is also shown in all figures.

It has been reported that PET generates a large amount of carbonaceous residue in a nitrogen atmosphere [60]. Furthermore, it is noted that the weight of residue for all the composites that were heated under oxidative-degradation conditions is lower than under a nitrogen atmosphere. This could be attributed to high temperature (460–580 °C) where oxygen reacts with carbon causing further weight loss of composites. Moreover, all the PET/carbon composites under nitrogen exhibit one decomposition step at ∼380 °C, whereas in air they show two decomposition steps. The first step is due to the degradation of PET chains and the second one is associated with a thermal degradation of char products that were formed during the first decomposition step. Similar behaviors have been reported in the literature [61] for PET/EG nano-composites and PET/clay nano-composites systems.

The $T_{5\%}$ of pure PET are ∼382 °C and 370 °C, in nitrogen and air, respectively, and increase to ≈392 °C and ≈384 °C upon 15 wt.% addition of the graphite. Moreover, the residual weight of PET is ≈9.5 wt.% at about 580 °C in nitrogen, while no residual weight was observed in air. In addition, the amount of residue increases with an increasing graphite content. In general, it has been established that the thermal stability of PET/graphite micro-composites is increased (4–9 °C) in nitrogen and (8–18 °C) in air atmospheres, compared to a pure PET. The $T_{5\%}$value of 368 °C recorded at 5 wt.% graphite differs from rest of the loadings. For the time being, we have to record this as an anomaly.

Graphite has been used in previous studies for the enhancement of thermal stabilities of several polymer matrices. For example, Otieno et al. [62] studied the thermal behavior of PU/graphite composites, and reported enhancement of thermal stability by about 20 °C at 50 wt.% loading. The thermal stability of a HDPE/graphite composite was reported by Wang et al. [21] who found an improvement of only 3 °C with increasing graphite contents up to 50 wt.%. Moreover, an addition of 5 wt.% of graphite was shown to give an enhancement of the $T_{5\%}$ of an epoxy resin [26] by 25 °C under nitrogen.

## 4. Mechanical Characterization and Modeling of Filled Polymers

### 4.1. Mechanical Characterization

We conducted as a series of mechanical experiments on the graphite-filled PET composite for two reasons. First, an experimental study on the filled PET with tensile tests can provide a good understanding on its mechanical properties. Secondly, these experimental results facilitate developing mathematical models for a computer-based simulation for predicting some material behaviors. For these, specimens for tensile testing were prepared using Haake Minijet II injection molding machine (Thermo Electron Corp., Hamburg, Germany). The tensile testing was performed using an Instron instrument of model 4301 at crosshead speed of 5 mm/min by following ASTM D638 standard. To ensure accuracy and reproducibility, five tests were performed and the average of them was reported as the final value of tensile properties. All the tensile tests were carried out at room temperature.

As expected, a rigid micro-filler-like graphite increases the composite stiffness, which is reflected in Figure 6 (top) by a continuous increment of the composite modulus of elasticity. In Figure 6 (bottom), a comparison is presented with some data available in the literature. Afterwards, several uniaxial-type tensile tests have been performed for various filler concentrations. The stress–strain graph presented in Figure 7 illustrates that, in addition to the stiffness enhancement by the graphite filler, it increases the embrittlement of the PET/graphite composite in contrast to a pure PET. Therefore, a PET/graphite composite with 15 wt.% filler breaks at a tensile strain of 1.5% compared to its pure counterpart that breaks at a deformation of 6.5%.

### 4.2. Constitutive Modeling of Filled Polymer

The understanding of mechanical behavior of filler-reinforced polymer has been an active field of research for last several decades. Particularly, the constitutive modeling to predict mechanical responses under the influence of micro-scale fillers have increasingly been studied [32,33,34]. Most of the earlier filled polymer modeling works were concerned with small deformations [32,33]. For large deformations, some advanced models, which are based on the so-called finite strain theory, can be found in the literature [34]. Moreover, computer-based numerical simulations have been performed in order to gain in-depth understanding of the mechanical responses of particle-filled composites.

#### 4.2.1. Small Strain-Based Particle-Filled Polymer Models

For small strain-based problems, the prediction of the equilibrium behavior is considered as a composite theory problem that can be approached in two different ways: one can derive rigorous bounds for the behavior or one can try to estimate the overall behavior. For the second case, one of the earliest and simplest approaches was due to Guth [32] and Guth and Gold [33], wherein the prediction of one of the most important linear elastic constants, i.e., Young’s modulus of a particle-filled solid, is expressed as $E_{c}=E_{m}\left(1+2.5v_{f}\right)$. Note that $E_{c},E_{m}$ are the Young’s moduli of the composite (*c*) and the matrix (*m*), respectively, and $v_{f}$ is the weight fraction of filler (expressed in percentage). According to the infinitesimal (small strain) theory, this estimate however is only good at very low filler concentrations and small deformation ranges. Therefore, several attempts have been made to improve model predictions by adding more terms to a polynomial series expansion of the amplification factor such as
(4)$$\begin{array}{c} E_{c}=E_{m}\left[1+2.5v_{f}+14.1v_{f}^{2}\right],\;E_{c}=E_{m}\left[1+0.67gv_{f}+1.62{\left[gv_{f}\right]}^{2}\right], \end{array}$$
where the shape factor *g* is related to length/breadth of a cluster and its value is assumed larger than 1. There are small-strain based material models for composites that can be easily be incorporated. However, our target is to model the experimental data presented in Figure 7 where deformations are far more than one percent (>1%).

#### 4.2.2. Large Strain-Based Particle-Filled Polymer Models

Most of the small strain-inspired models are based on the so-called mixture theories, which have taken some sort of account of the complex behaviors of filler–matrix interactions, e.g., particle size, shape, orientation, and interaction between particle and matrix. However, finite strain based models develop mainly around the concept of the amplification factor in a homogenized sense. For modeling polymeric materials at large deformations, the starting point is the deformation gradient F that represents large strain deformations where the left Cauchy–Green strain tensor is defined as $b=FF^{T}$. In general, polymeric materials are considered as incompressible or nearly incompressible solids which can be characterized by an strain energy function as
(5)$$\begin{array}{c} \Psi =\tilde{\Psi }\left(I_{1},I_{2},I_{3}\right). \end{array}$$

In Equation (Equation 5), the three strain invariants $I_{1},I_{2},I_{3}$ are defined as $I_{1}=\lambda_{1}^{2}+\lambda_{2}^{2}+\lambda_{3}^{2};$$I_{2}=\lambda_{1}^{2}\lambda_{2}^{2}+\lambda_{2}^{2}\lambda_{3}^{2}+\lambda_{1}^{2}\lambda_{3}^{2}$; $I_{3}=\lambda_{1}^{2}\lambda_{2}^{2}\lambda_{3}^{2}$, where $\lambda_{1},\lambda_{2},\lambda_{3}$ are the three eigenvalues of the deformation tensor b. The first eigenvalue $\lambda_{1}$ represents the stretch in the *x* direction while $\lambda_{2}$ and $\lambda_{2}$ denote stretches in the *y* and *z* directions, respectively. Note that for the incompressible materials, the third invariant $I_{3}$ becomes unity, i.e., $I_{3}=1$. Henceforth, the Cauchy stress tensor (S) may be expressed as
(6)$$\begin{array}{c} S=-pI+2\frac{\partial \Psi }{\partial I_{1}}b-2\frac{\partial \Psi }{\partial I_{2}}b^{-1}, \end{array}$$
where *p* is a scalar quantity that serves as an indeterminate Lagrange multiplier and will be removed from the above equation using appropriate boundary conditions. In Equation (Equation 6), $b^{-1}$ is the inverse of the deformation tensor b and I is a second order identity tensor. We conducted all experiments described in the previous sections that fulfill usual dimensions of a uniaxial deformation where the deformation gradient and the left Cauchy–Green tensor can be defined as
(7)$$F=\left[\begin{array}{ccc} \lambda_{1} & 0 & 0\\ 0 & \lambda_{2}^{-1/2} & 0\\ 0 & 0 & \lambda_{3}^{-1/2} \end{array}\right]=\left[\begin{array}{ccc} \lambda & 0 & 0\\ 0 & \lambda^{-1/2} & 0\\ 0 & 0 & \lambda^{-1/2} \end{array}\right];\;b=FF^{T}=\left[\begin{array}{ccc} \lambda^{2} & 0 & 0\\ 0 & \lambda^{-1} & 0\\ 0 & 0 & \lambda^{-1} \end{array}\right].$$

In deriving the above equation, the condition of incompressibility is applied, i.e., $\lambda_{1}=\lambda ,$$\lambda_{2}=\lambda_{3}=1/\sqrt{\lambda }\left(\lambda_{1}\lambda_{2}\lambda_{3}=1\right)$. Note that λ is called stretch, which is commonly used in large deformation theories, while ε is termed as a strain and they are related by λ=ε+1. To predict the behavior of unfilled polymers, at first we take energy functions of two widely-used classical models, i.e., neo-Hooke (NH) and Carrol (C) models as
(8)$$\begin{array}{c} \Psi_{\mathrm{NH}}=\frac{\mu }{2}\left(I_{1}-3\right),\;\Psi_{C}=aI_{1}+bI_{1}^{4}+c\sqrt{I_{2}} \end{array}$$
where μ,a,b,c are material parameters of respective models that need to be identified from appropriate sets of experimental data (see Steinmann et al. [63]). Note that there are two modeling approaches for polymers discussed in the literature [63,64,65,66]: (i) micro-mechanically motivated models; and (ii) phenomenological-motivated models. The neo-Hookean model we used initially has both the micro-mechanical and phenomenological explanations [34,66,67]. Our modeling approach is not a “curve-fitting” work since the energy function used here is based on three-dimensional framework that has been formulated obeying some basic principles of thermodynamics [34,67]. It starts from a general description of the problem and then we gradually derive the one-dimensional version of the stress–strain relation since our experimental works are only for uniaxial tests. Polynomial type “curve-fitting” models cannot be extended to any three-dimensional Finite Element simulations, which is our ultimate target of material modeling. Applying the differentiation defined in Equation (Equation 6), the uniaxial stress expressions for unfilled polymers with the two models can be derived as
(9)$$\begin{array}{c} S_{NH}=\mu \left[\lambda^{2}-\lambda^{-1}\right],\;S_{C}=\left[2a+8b{\left[2\lambda^{-1}+\lambda^{2}\right]}^{3}+c{\left[1+2\lambda^{3}\right]}^{-\frac{1}{2}}\right]\left[\lambda^{2}-\lambda^{-1}\right] \end{array}$$

For detailed derivations of Equation (Equation 9), readers may consult the works of Steinmann et al. [63], Hossain et al. [64,65,68], and Liao et al. [69]. To predict the behavior of filled polymers at large strains, Mullins and Tobin [70] introduced the notion of so-called amplified stretch Λ, which in the case of a uniaxial loading, is related to the actual axial stretch λ by
(10)Λ=1+X(λ−1)
where *X* is the stretch amplification factor that depends on the fraction of filler $v_{f}$. For instance, according to Guth model, this factor can be defined as
(11)$$X=1+0.67gv_{f}+1.62{\left[gv_{f}\right]}^{2}$$
where $v_{f}$ is the weight fraction of fillers and *g* is a factor describing the asymmetric nature of the aggregated clusters. Therefore, to obtain complete stress–stretch expressions, according to Mullins–Tubin assumption, the actual stretch λ will be replaced by the amplified stretch Λ
(12)$$\begin{array}{c} S_{NH}=\mu \left[\Lambda^{2}-\Lambda^{-1}\right],\;S_{C}=\left[2a+8b{\left[2\Lambda^{-1}+\Lambda^{2}\right]}^{3}+c{\left[1+2\Lambda^{3}\right]}^{-\frac{1}{2}}\right]\left[\Lambda^{2}-\Lambda^{-1}\right] \end{array}$$

Note that, in our studies, we have calculated the uniaxial engineering stress (or nominal stress that is calculated as the total force over the original sample cross-sectional area) *P* which relates to the uniaxial Cauchy stress via S=λP.

### 4.3. Parameter Identification and Model Validation

#### 4.3.1. Linear Models

As shown in the previous section, t fillers increase the mechanical stiffness of the reinforced-PET composites, which is depicted with increments of the elastic moduli (see Figure 5). At first, the model for elastic moduli enhanced presented in Equation ${\left(4\right)}_{2}$ is fitted to the experimental data. Results are presented in Figure 8 (top) where the parameter *g* is identified as 2.67, which is within the range (larger than one).

After the parameter identification within elastic enhancement model in Equation (Equation 4), first, the NH linear model was selected to validate the stress–stretch experimental data. Note that the relation between the composite elastic modulus $E_{c}$ and the parameter μ appearing in the NH model is related to $\mu =\frac{E_{c}}{2[1+\nu ]}$, where ν is the Poisson’s ratio that is taken as 0.5 for the incompressible polymeric material under consideration. The model prediction with the linear NH model overestimates the experimental results. It may be the reason that the deformation range is more than six percent in our experiments, which illustrates a predominate nonlinearity in the stress–stretch curves (see Figure 8, bottom). Hence, a linear model such as NH with only one material constant fails to predict the experimental data.

#### 4.3.2. Non-Linear Models

Next, a nonlinear model, i.e., the Carrol model presented in Equation ${\left(12\right)}_{2}$, was used to fit the experimental data of the pure (unfilled) PET that breaks at a 6.5% deformation. Figure 9 (top) presents the model fitting with experimental data of the unfilled PET (circle dots are for the unfilled PET) with the identified parameters as [a,b,c] = [1.44e + 4 MPa, −112.6 MPa, 7493 MPa]. Once Carrol model parameter is identified with the help of the unfilled PET data, the model needs to be validated with other data that are not used in the parameter identification. For the model validation with the filled PET data, the shape parameter *g* appearing in the amplification factor *X* needs to be identified first. In this case, we used the value 2.67 already identified by the linear model of modulus of elasticity enhancements (data in Figure 6, top) using $E_{c}=E_{m}\left[1+0.67gv_{f}+1.62{\left[gv_{f}\right]}^{2}\right]$. Predictions of the stress–stretch behaviors of four different PET/graphite materials are presented in Figure 9. All predictions result in good agreements with the experimental data.

## 5. Conclusions

In this study, conductive graphite filled PET micro-composites were prepared from PET as a matrix and graphite as a micro-filler by a melt compounding technique using a co-rotating twin-screw extruder and molded into films and tensile test pieces by compression and injection moldings, respectively. The electrical conductivity, morphology, thermal stability, crystallization rate, and degree of crystallinity of these composites were studied and characterized. In particular, at ∼14.5 wt.% of graphite, a significant enhancement in the electrical conductivity (σ) was observed. Such loading marks the insulator–conductor transition, having a percolation threshold value ($\Phi_{c}$) of 14.7 wt.%. The electric conductivity (σ) was found to be around 0.0016 S/m at or just above the $\Phi_{c}$ value (14.7 wt.%). Such σ value is required for anti-static applications. In addition, a higher agglomeration and poorer distribution of the graphite were also observed, resulting in the establishment of electrical conductive network. The addition of graphite into the PET further caused an increase in the degree of crystallinity, accelerated both cold and melt crystallizations and improved the thermal stability. It is further revealed that a large deformation-based material model can successfully predict the mechanical responses of the graphite enhanced PET/graphite composite polymers.

## Figures and Tables

**Figure 1 polymers-11-01411-f001:**
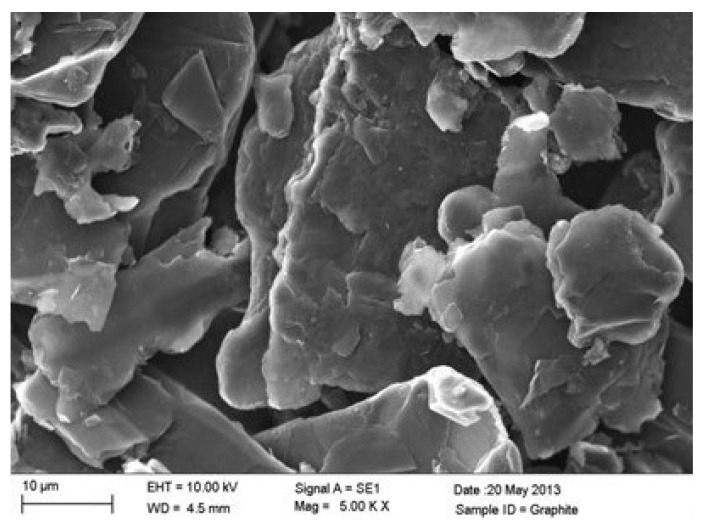
SEM images of as received-graphite.

**Figure 2 polymers-11-01411-f002:**
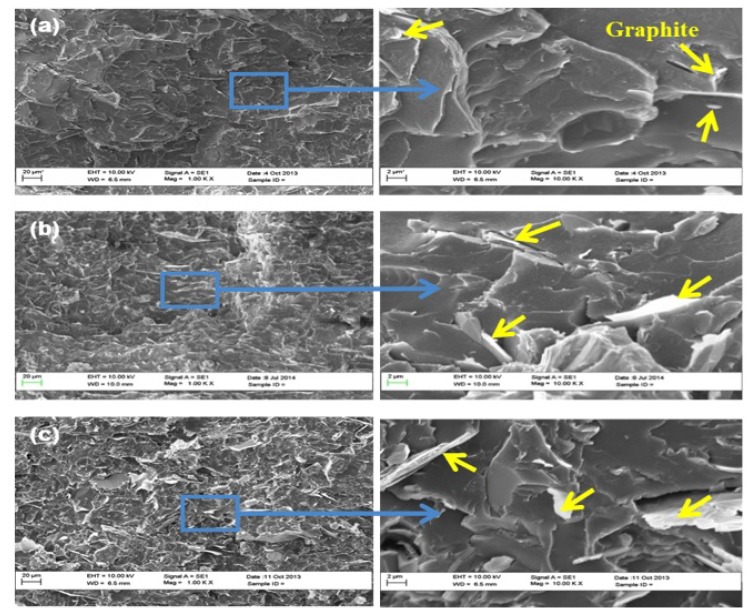
SEM images of PET/graphite micro-composites at low (**left**) and high magnifications (**right**): (**a**) 10 wt.%; (**b**) 15 wt.%; and (**c**) 20 wt.% of graphite. The arrows in the magnified images indicate the graphite.

**Figure 3 polymers-11-01411-f003:**
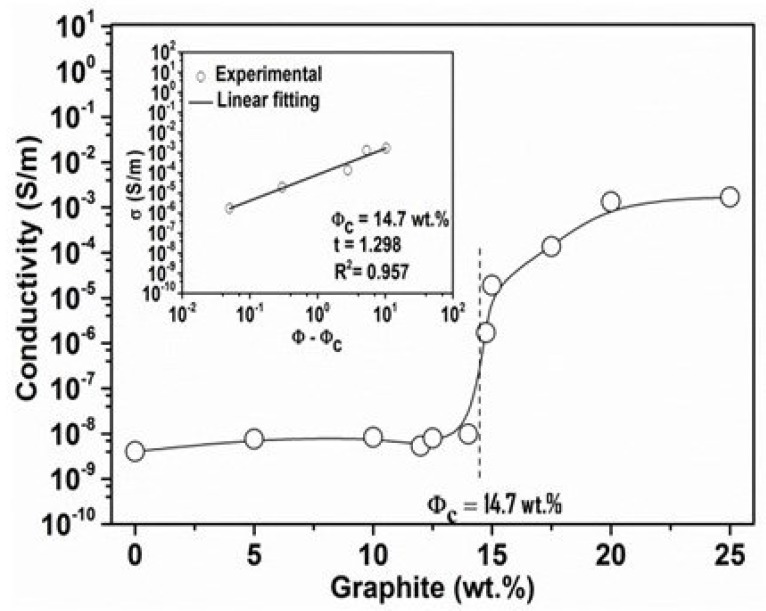
Percolation threshold ($\Phi_{c}$) determination plot. Electrical conductivity of PET/graphite micro-composites as a function of graphite content. The inset plot is logσ versus $\log(\Phi -\Phi_{c})$.

**Figure 4 polymers-11-01411-f004:**
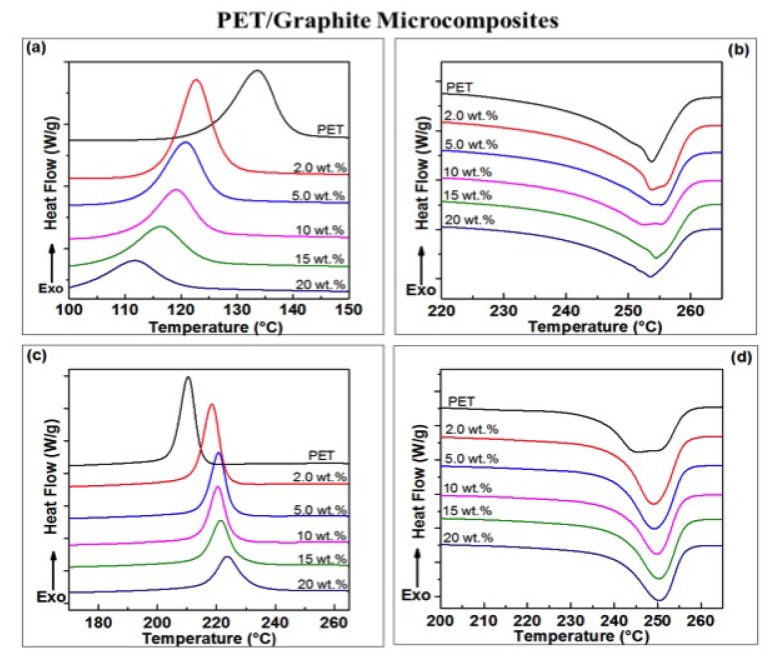
DSC curves (heating/cooling rate 10 °C/min) for PET/graphite micro-composites show: (**a**) cold crystallization peaks; (**b**) melting peaks from the first scan; (**c**) subsequent cooling curves; and (**d**) melting peaks from the second heating scan.

**Figure 5 polymers-11-01411-f005:**
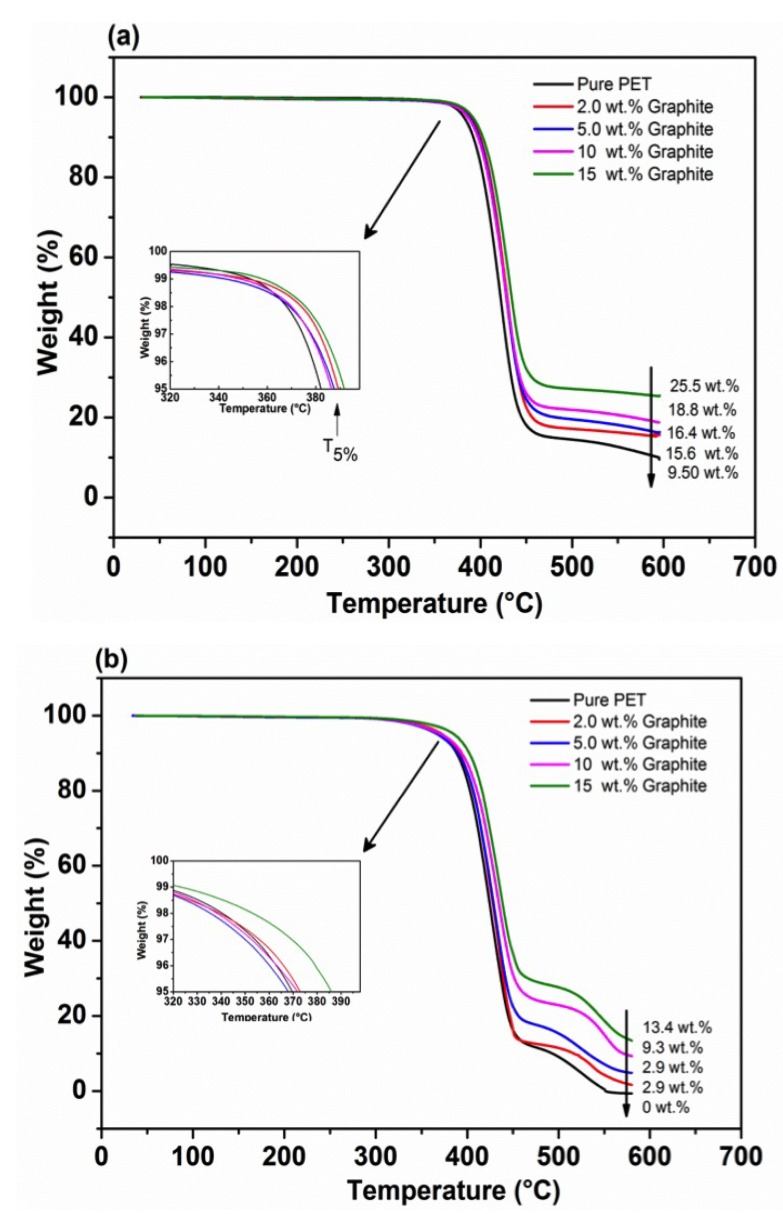
TGA thermograms (heating rate of 10 °C/min) of PET/graphite micro-composites with various graphite contents examined under: (**a**) nitrogen atmosphere; and (**b**) air atmosphere.

**Figure 6 polymers-11-01411-f006:**
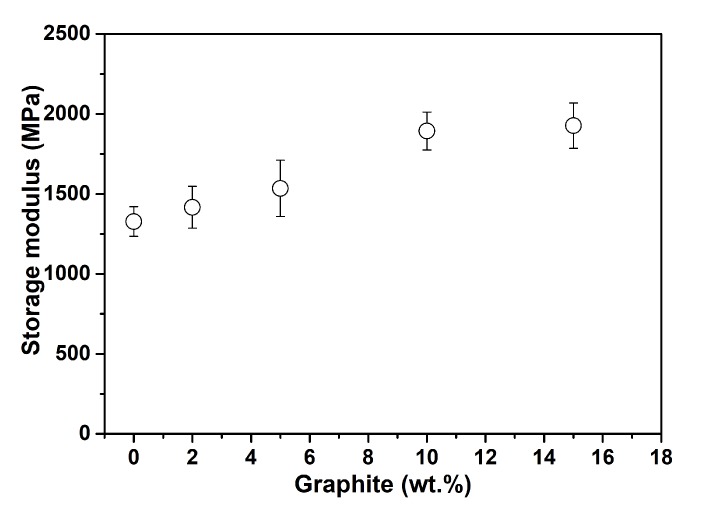
Modulus of elasticity enhancements of the PET/graphite composite: (**top**) modulus enhancements with respect to various filler concentrations in our study; and (**bottom**) our study and literature values, a comparison.

**Figure 7 polymers-11-01411-f007:**
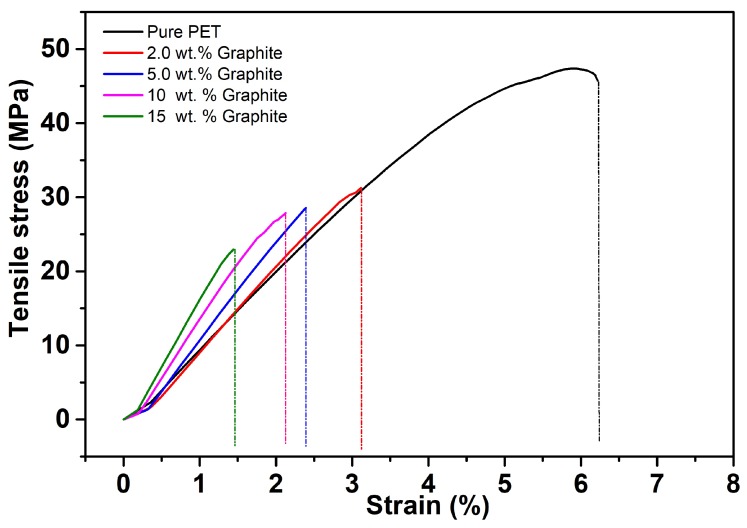
Stress–strain responses with respect to various graphite concentrations. As expected, the rigid graphite filler stiffens the PET matrix and reduces its deformability.

**Figure 8 polymers-11-01411-f008:**
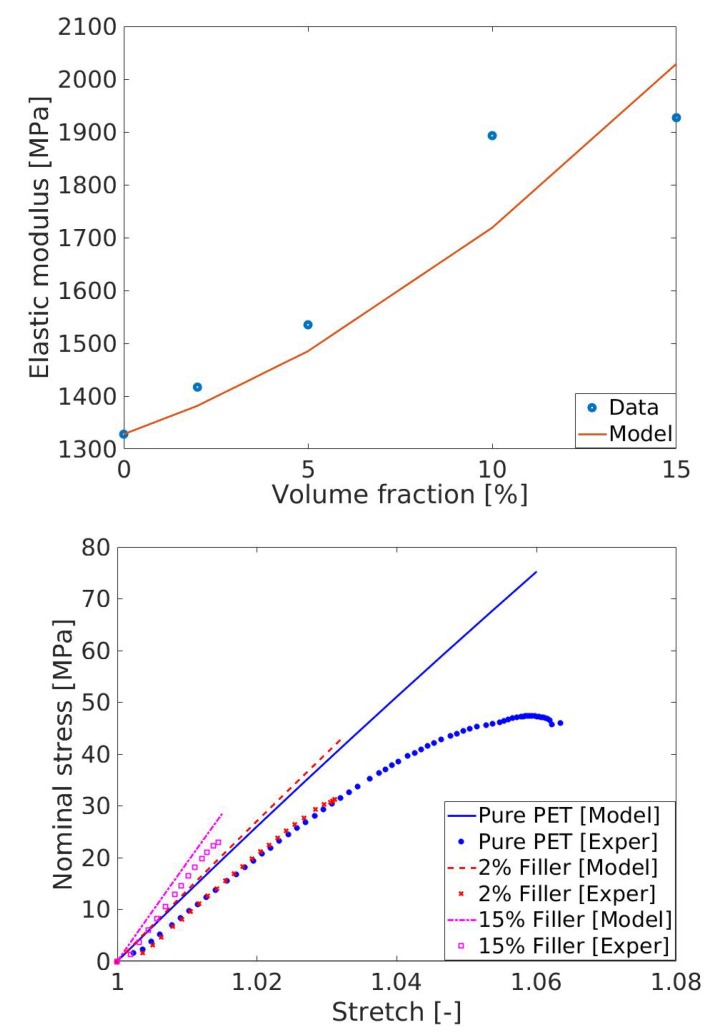
(**Top**) *Model fitting* of elastic moduli at various fraction of fillers; and (**Bottom**) *model prediction* with the linear Neo-Hooke model. NH model yields unsatisfactory fitting with the experimental data.

**Figure 9 polymers-11-01411-f009:**
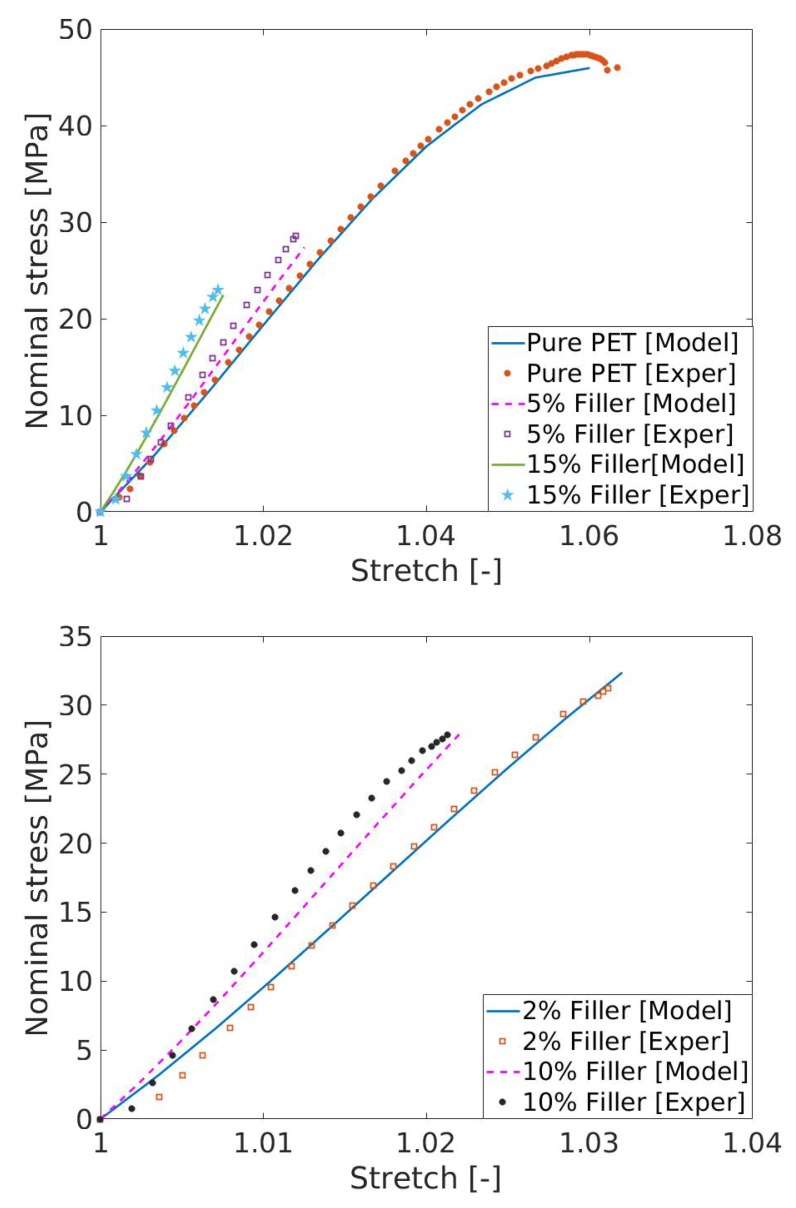
(**Top**) *Model prediction* with the nonlinear Carrol model 5% and 15% fillers; and (**Bottom**) *model prediction* with the nonlinear Carrol model for 2% and 10%.

**Table 1 polymers-11-01411-t001:** DSC data for PET/graphite micro-composites.

Graphite (wt.%)	$T_{g}\;\left({}^{\circ}C\right)$	$T_{\mathrm{cc}}\;\left({}^{\circ}C\right)$	$T_{m}\;{\left({}^{\circ}C\right)}^{\;*}$	$T_{m}\;{\left({}^{\circ}C\right)}^{\;**}$	$T_{\mathrm{mc}}\;\left({}^{\circ}C\right)$	$X_{c}\;{\left(\%\right)}^{\;***}$
0	72.9 ± 1.2	133.5 ± 0.1	253.4 ± 0.4	245.9 ± 0.9	209.2 ± 0.1	11.8 ± 1.8
2	73.9 ± 0.1	122.6 ± 0.5	254.4 ± 0.9	249.1 ± 0.1	218.1 ± 0.8	15.9 ± 2.0
5	74.2 ± 0.1	120.5 ± 1.5	254.9 ± 1.0	249.2 ± 0.4	220.5 ± 0.5	15.6 ± 1.6
10	73.8 ± 0.1	119.1 ± 0.5	253.5 ± 1.7	249.8 ± 0.2	220.1 ± 0.4	15.8 ± 2.0
15	65.0 ± 0.5	116.5 ± 0.1	253.7 ± 1.0	249.9 ± 0.4	221.2 ± 0.2	19.8 ± 1.1
20	64.1 ± 0.1	111.6 ± 0.7	253.0 ± 0.9	250.5 ± 0.2	223.4 ± 0.1	21.3 ± 2.8

${}^{*}$ Melting temperatures obtained from the first heating scans, ${}^{**}$ Melting temperatures obtained from the second heating scans, ${}^{***}$ Crystallinity after quenching ($X_{c}=X_{m}-X_{cc}$), where $X_{m}$ is the degree of crystallinity associated with melting processes and $X_{cc}$ is the degree of crystallinity associated with cold crystallization process.

**Table 2 polymers-11-01411-t002:** Thermal degradation temperature ($T_{5\%}$) for PET/graphite micro-composites under nitrogen and air atmospheres.

Graphite (wt.%)	$N_{2}$	Air
0	382.3 ± 1.4	370.1 ± 1.0
2	389.8 ± 0.8	373.1 ± 6.0
5	387.7 ± 0.4	368.0 ± 4.3
10	386.7 ± 3.0	372.7 ± 2.6
15	391.7 ± 0.5	383.7 ± 1.9

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
