# Peer review of "Addition of Graphite Filler to Enhance Electrical, Morphological, Thermal, and Mechanical Properties in Poly (Ethylene Terephthalate): Experimental Characterization and Material Modeling"

_polymers, 2019, doi:10.3390/polym11091411_

Round 1
Reviewer 1 Report
Please check my attachment.

Author Response
Dear editor,
We appreciate you for giving us the opportunity to revise our manuscript with some corrections. We agree with the correction of the reviewers and have tried our earnest to correct our manuscript as suggested. We have thoroughly checked our manuscript for grammatical mistakes. The changes are marked as dark blue color in the manuscript and also given below. We would like to receive final decision on our manuscript.
Please feel free to contact me if you need anything regarding the publication process of our manuscript.
With regards,
Mohammad Rezaul Karim, PhD
Associate Professor
King Saud University
Riyadh, KSA
e-mail: mkarim@ksu.edu.sa
Phone: +966551460354
Reviewers suggested corrections
Reviewer #1: Comments
In this work, authors summarized some result of characterization for the PET/graphite composite. Some mechanical tests were conducted, and some modeling work is presented. Generally, the PET/graphite composite system is well studied. No new experimental method or good property is pointed out by this work. Not much new knowledge is introduced either. To improve this work, at least find some new phenomenon or use some new method or enhance some property. Some mistakes or misleading and my concerns in this manuscript are mentioned as the followings:
Suggested corrections
As statement in line 85 and 86, authors think the mixing method by extrusion is a new idea. But as my knowledge, this is one of the most common method to mix the filler and polymer matrix. For the PET/graphite system, many papers are reported by this method, such as JPS PART B, 2012, 50, 1645- 1652 (single extruder) and Composites part A, 2011, 42, 560-566 (twin extruder).
Ans.: We respect reviewer’s comments. It has been corrected.
In line 118, why PET can be a resin? Please check the meaning of term “resin”. Moreover, the basic property of PET is not included such the molecular weight.
Ans.: Yes, it is a thermoplastic polymer resin.
For the figure 2, it seems like that authors consider the bright edge is one indication for the graphite. There is no reason for this. The edge of polymer crystal or aggregates also may also charge and show brightness. Based on the morphology and the imagination, some placed pointed out by authors are needed to reconsider.
Ans.: It has been corrected.
For the property of conducting, percolation threshold is 14.7%. This is a high value for the loading filler. Some other filler is much more conducting than this such as graphene.
Ans.: The percolation threshold is dependent on the matrix used and the method employed to prepare the composite material. Krupa et al. [18] examined electrical properties of PE/graphite composites and reported even higher percolation threshold value of 11 vol. % (24 wt.%). In another study, She et al. [40] reported also higher percolation threshold values, i.e. 22.2 wt.% for PE/graphite micro-composites. Therefore, the values available in the literature are higher in comparison to those obtained in the present study. The difference in percolation threshold values could be attributed to the difference in filler sizes, matrices and preparation method used.
The discussion of DSC is needed to be clarified. The first and the second cycle are exhibiting different manners and authors chose first cycle to do the calculation and discussion. Do the authors want to study the history of fabrication or the property of composite? Unless some study focuses on the process of manufacturing, the second cycle should be used. Because the samples were quenched very fast, so the first cycle normally can’t reflect the property of the composite. The discussion and the conclusion should reconsider. For the conclusion of the crystallization and nucleation, the on-set point is only one type of indication. The iso-thermal crystallization test is needed.
Ans.: The % degree of crystallinity (Xc) of PET and its graphite composite was calculated from the first-heat data as these data reflect the thermal history of the PET/carbon composites. But we run the second heating which is after removing history effect of these composites, recording of DSC curves after removing of its thermal history for all samples and we noticed that No cold crystallization is visible and both samples show a double melting peak. This double peak behavior is a well-known phenomenon for many semi-crystalline polymers.
For the TGA test, 5% weight loss is chosen as the position for comparison. Commonly, 2% or 3% is a good value for this. So, the enhancement is over estimated for this study.
Ans.: Thermal stability is commonly evaluated by determining the onset of thermal decomposition temperature (IDT) and the temperature at 0.05 (5%) weight loss (T5%).
For the mechanical test, authors need to understand basic knowledge of mechanical terms such as what is young’s modulus and what is storage modulus. What is stress-strain curve? Some terms are misleading such as what does stress-stretch mean? Is it displacement or strain? For the Figure 6 bottom, what is the meaning for it? Does this mean some other work is better than the material in this study?
Ans.: We appreciate reviewer’s comments. Definitely, there are some procedural differences for calculating Young’s modulus and storage modulus. With DMA, the so-called storage (E’) and loss moduli (E”) can be obtained [Ref: Mechanics of Solid Polymers, J. Bergstroem, 2016]. From there, the elastic modulus/Young’s modulus can be called as . Since the materials, we are dealing with here, are less dissipative (E” is almost zero). The storage modulus can easily be considered equal to the Young’s modulus. ‘Stretch’ ( ) is a deformation where ( is a ‘strain’, we mentioned in the manuscript). For modelling in the large deformation, strain ( ) is usually termed as stretch( ). Hence, to be consistent with the literature, we used the term ‘stretch’ instead of strain.
For the model part, not much new knowledge is introduced. No physical meaning is discussed. If only the adjusting is math, any curve can be described by polynomial. More terms of polynomial certainly can close the date to the curve. The model needs new idea or understanding.
Ans.: As per suggestion, we discuss more on the physical meaning of the models used in the current study. Our modelling approach is not ‘curve-fitting’ work since the energy function is based on some basic principles of thermodynamics [66, 69, 70, 74,75]. It starts from a general description of the problem and then we gradually derive the one-dimensional version of the stress-strain relation since experimental works presented here are only for uniaxial tests. Polynomial type ‘curve-fitting’ models cannot be extended to any three-dimensional Finite Element simulations, which is our next target.

Reviewer 2 Report
Dear Author(s),
The manuscript Polymers-548087 and titled as “Addition of graphite filler to enhance electrical, morphological, thermal, and mechanical properties in poly (ethylene terephthalate): Experimental characterization and material modeling” was reviewed. It is quite interesting paper. I accept the manuscript after following revisions are made.
1. Please reconsider the your propose small strain and non-hookanian models and do relations with traditional fiber based laminate composite (if any). Non-hookanian model look likes parametrical polynominal relations in which it is hardly reliable.
My best regards,
Author Response
Reviewer #2: Comments
The manuscript Polymers-548087 and titled as “Addition of graphite filler to enhance electrical, morphological, thermal, and mechanical properties in poly (ethylene terephthalate): Experimental characterization and material modeling” was reviewed. It is quite interesting paper. I accept the manuscript after following revisions are made.
Please reconsider the your propose small strain and non-Hookean models and do relations with traditional fiber based laminate composite (if any). Non-Hookean model look likes parametrical polynominal relations in which it is hardly reliable.
Ans.: We would like to appreciate comments from the reviewer. There are small-strain based material models for composites that can be easily be incorporated. However, our target is to model the experimental data presented in Fig 7 where deformations are far more than one percent (> 1% ). Hence, we need to go for a material model that is based on large deformations. In this case, there are two approaches discussed in the literature [66, 69,70, 74] i) micro-mechanically motivated models and ii) phenomenological-motivated models. The neo-Hookean model (e.g., a non-Hookean model), we used initially, has both the micro-mechanical and phenomenological explanations [66, 69, 75]. Our modelling approach is not ‘curve-fitting’ work since the energy function used here is based on three-dimensional framework that has been formulated obeying some basic principles of thermodynamics [66, 75]. It starts from a general description of the problem and then we gradually derive the one-dimensional version of the stress-strain relation since our experimental works are only for uniaxial tests. Polynomial type ‘curve-fitting’ models cannot be extended to any three-dimensional Finite Element simulations, which is our ultimate target of material modelling.

Reviewer 3 Report
In this paper, experimental characterization and material modelling of PET/graphite micro-composites have been investigated. Although this work has been well written and some valuable results have been obtained, there are still some problems that should be clarified before further consideration.
1. The resolution of Figure 1, 2, 3, 4 and 5 are not good. Please improve them.
2. In introduction, the authors stated that “earlier studies have been demonstrated that even low addition levels of nano-fillers can give significant improvements in the electrical, mechanical, and thermal properties of polymers.” (line 28-30). In fact, the nano-fillers can also improve the flame retardant properties of polymers. Please refer to the following articles: 1) ACS applied materials & interfaces, 2016, 8(39): 26266-26274; 2) Nanomaterials, 2018, 8(2): 70.
3. Can the authors provide the thickness of graphite?
4. In DSC characterization, there is no statement of eliminating the thermal history. For example, samples should be heated from room temperature to 270 °C at 10 °C/min and held at 270 °C for 5 min to eliminate the thermal history, then cooled to room temperature at 10 °C/min. The thermal history may influence the crystallization temperature. Please clarify.
Author Response
Reviewer #3 Comments:
In this paper, experimental characterization and material modelling of PET/graphite micro-composites have been investigated. Although this work has been well written and some valuable results have been obtained, there are still some problems that should be clarified before further consideration.
The resolution of Figure 1, 2, 3, 4 and 5 are not good. Please improve them.Ans.: The figures 1-5 have been changed.
In introduction, the authors stated that “earlier studies have been demonstrated that even low addition levels of nano-fillers can give significant improvements in the electrical, mechanical, and thermal properties of polymers.” (line 28-30). In fact, the nano-fillers can also improve the flame retardant properties of polymers. Please refer to the following articles: 1) ACS applied materials & interfaces, 2016, 8(39): 26266-26274; 2) Nanomaterials, 2018, 8(2): 70.Ans. We would like to thank the reviewer for mentioning some important references. We have added all relevant references.
Can the authors provide the thickness of graphite?Ans: As-received flakes of synthetic graphite were in the form of powder of particle size < 20 µm.
In DSC characterization, there is no statement of eliminating the thermal history. For example, samples should be heated from room temperature to 270 °C at 10 °C/min and held at 270 °C for 5 min to eliminate the thermal history, then cooled to room temperature at 10 °C/min. The thermal history may influence the crystallization temperature. Please clarify.Ans: The thermal history may affect the crystallization temperature; therefore the % degree of crystallinity (Xc) of PET and its carbon composite was calculated from the first-heat data as these data reflect the thermal history of the PET/carbon composites. DSC curves show a range of peaks; cold crystallization peaks, melting peaks from the first heating scans, melt crystallization peaks during cooling. To erase their thermal history, samples were kept at 270 °C for 5 minutes and then cooled down to 23 °C the melting peaks from second heating scans of the PET/graphite micro composites.

Round 2
Reviewer 1 Report
Thank you very much for the authors' correction and considering my suggestion. I only have one more small question,
For my second question, the term "resin" normally is describing one polymer mixture which the structure is unknown. Such as epoxy resin means the how many repeat union is there is not sure and it may mixed with some molecules with unknown structure.
For the structure of PET which is well known, normally the molecular weight is reported. If the term resin is used, for the polymer scientist may misunderstand this polymer has certain uncertain structure in it.
Author Response
We respect the reviewer's comments. It has been updated as per reviewer's suggestion.